

# Heat stress does not induce wasting symptoms in the giant California sea cucumber (*Apostichopus californicus*)

Declan Dawson Taylor[1,2,*], Jonathan J. Farr[2,3,*], Em G. Lim[4], Jenna L. Fleet[2,5], Sara J. Smith[2,6,7,8] and Daniel M. Wuitchik[2,6]

[1] Earth, Ocean and Atmospheric Sciences, University of British Columbia, Vancouver, British Columbia, Canada
[2] Bamfield Marine Sciences Center, Bamfield, British Columbia, Canada
[3] Biological Sciences, University of Alberta, Edmonton, Alberta, Canada
[4] Biological Sciences, Simon Fraser University, Vancouver, British Columbia, Canada
[5] Biological Sciences, University of Winnipeg, Winnipeg, MB, Canada
[6] Biological Sciences, Boston University, Boston, MA, United States of America
[7] Informatics Group, Harvard University, Cambridge, MA, United States of America
[8] Biology, Mount Royal University, Calgary, Alberta, Canada
[*] These authors contributed equally to this work.

Corresponding authors
Sara J. Smith,
sara.wuitchik@gmail.com,
ssmith6@mtroyal.ca
Daniel M. Wuitchik,
wuitchik@bu.edu

## ABSTRACT

Oceanic heatwaves have significant impacts on disease dynamics in marine ecosystems. Following an extreme heatwave in Nanoose Bay, British Columbia, Canada, a severe sea cucumber wasting event occurred that resulted in the mass mortality of *Apostichopus californicus*. Here, we sought to determine if heat stress in isolation could trigger wasting symptoms in *A. californicus*. We exposed sea cucumbers to (i) a simulated marine heatwave (22 °C), (ii) an elevated temperature treatment (17 °C), or (iii) control conditions (12 °C). We measured the presence of skin lesions, mortality, posture maintenance, antipredator defences, spawning, and organ evisceration during the 79-hour thermal exposure, as well as 7-days post-exposure. Both the 22 °C and 17 °C treatments elicited stress responses where individuals exhibited a reduced ability to maintain posture and an increase in stress spawning. The 22 °C heatwave was particularly stressful, as it was the only treatment where mortality was observed. However, none of the treatments induced wasting symptoms as observed in the Nanoose Bay event. This study provides evidence that sea cucumber wasting may not be triggered by heat stress in isolation, leaving the cause of the mass mortality event observed in Nanoose unknown.

## INTRODUCTION

Climate change is increasing the intensity, duration, range, and frequency of marine heatwaves across the globe with potentially catastrophic effects on organism fitness, ecosystems, and human economies (*Frölicher, Fischer & Gruber, 2018*; *Allan et al., 2021*). In marine ecosystems these extreme climatic events often cause immediate and mass mortality at all trophic levels from thermal stress, starvation, toxicity, and hypoxia (*Cavole*

*et al., 2016; Di Lorenzo & Mantua, 2016; von Biela et al., 2019; Suryan et al., 2021*). Marine heatwaves may exacerbate disease because thermal stress can compromise the immune response, and warmer temperatures can increase the virulence of pathogens (*Harvell et al., 1999; Marcogliese, 2008; Branco et al., 2012; Matozzo et al., 2012; Oliver et al., 2017*). This temperature-associated increase in virulence and infectivity has been associated with stimulating and intensifying sea star wasting disease to devastating effects (*Harvell et al., 1999; Bates, Hilton & Harley, 2009; Eisenlord et al., 2016; Hewson et al., 2018; Aquino et al., 2021*). A notorious example of the impacts of a wasting disease outbreak was the 2013-current sea star wasting epidemic, which has affected more than 20 sea star species in the Northeast Pacific Ocean over the last decade (*Hewson et al., 2018*). Sea star wasting disease encompasses a broad set of symptoms including twisted arms, lesions, deflation/loss of turgor, loss of arms, lack of grip strength in tube feet, and liquefaction (*Bates, Hilton & Harley, 2009; Menge et al., 2016; Hewson et al., 2018*). In most sea star species, the driving cause of wasting disease remains largely uncertain. It has often been assumed that wasting in echinoderms is driven by infectious agents (*Hewson et al., 2014; Bucci et al., 2017; Miner et al., 2018; Hewson, Johnson & Tibbetts, 2020*). Warm temperatures have also been linked to several mass mortality events (*Bates, Hilton & Harley, 2009; Eisenlord et al., 2016; Harvell et al., 2019*). Because of the population-level impacts on ecologically important species, understanding the pathogenic and environmental drivers of sea star wasting remains an area of active research (*Aalto et al., 2020; Aquino et al., 2021; Hewson, 2021*).

Wasting is not limited to sea stars and is an emerging concern across closely related taxa. For example, sea urchins have faced a variety of disease-linked mass mortality events and epizootics (*Feehan & Scheibling, 2014*). Several of these bacterial and amoebic diseases have been linked to warm water anomalies, from climate events or storms (*Sweet, 2020*). Red sea urchins (*Mesocentrotus franciscanus*) and purple sea urchins (*Strongylocentrotus purpuratus*) have suffered from "bald sea urchin disease" and "sea urchin wasting disease" along the Pacific coast of British Columbia (B.C.). Pathology of these diseases includes lesions to the body wall and a shortening or loss of spines, and both have been associated with mass mortality events (*Feehan & Scheibling, 2014; Sweet, 2020*). While these diseases are believed to have bacterial origins, a single bacterial strain has not been identified as the cause (*Sweet, 2020*). Disease-mediated mass mortality events in urchins and sea stars have caused trophic cascades and dramatic ecosystem shifts, highlighting the importance of understanding these ecological phenomena (*Schultz, Cloutier & Côté, 2016; Suryan et al., 2021; Traiger et al., 2022*). Recent evidence has emerged that wasting may occur in sea cucumbers as well, promoting further concerns about the impacts of marine diseases in shallow, near-shore ecosystems.

Since 2014, there have been reports of giant California sea cucumbers (*Apostichopus californicus*) with wasting symptoms similar to those of sea stars throughout their range in the northeast Pacific Ocean (*Hewson, Johnson & Tibbetts, 2020*). Sea cucumber wasting disease is poorly defined, with symptoms typically including non-focal lesions and fissures across the body wall, epidermal tissue sloughing, and rapid liquefaction (*Hewson, Johnson & Tibbetts, 2020; Fig. 1C*). Much like in sea stars, there are numerous potential causal agents that could elicit this suite of symptoms. Heat stress may be particularly relevant

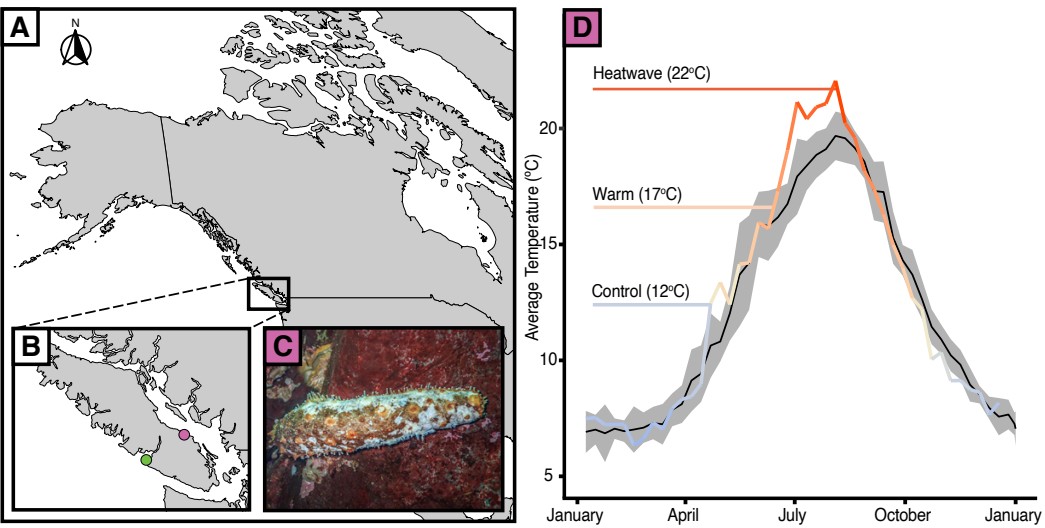

**Figure 1** **Temperature anomalies and sea cucumber wasting.** (A) Map of western North America with (B) an inset of Vancouver Island showing the data collection locations of Bamfield Inlet (green), Nanoose Bay (purple). (C) *A. californicus* in Nanoose Bay exhibiting wasting symptoms. (D) Eight-day rolling average seasonal sea surface temperatures in Nanoose Bay from 2010 to 2020 (black line with grey SE ribbon) and 2021 (colour gradient line), with experimental temperature treatments highlighted.

given that after a series of heatwaves from June 25 to July 07, 2021 (*Environment and Climate Change Canada, 2021*; Ocean Networks Canada) a wasting event with very high mortality occurred in the Strait of Georgia near Nanoose, B.C., from late August to October 2021 (Fig. 1D; E Lim, pers. comm., 2021). At its peak, up to 94% of observed *A. californicus* at a single site showed evidence of wasting. On average, across all affected sites ($n = 7$), 50% of observed *A. californicus* exhibited wasting symptoms. The six to ten week delay between peak near-surface sea temperatures and widespread mortality may be indicative of a potential link between thermal stress and wasting in *A. californicus*, perhaps mediated by the progression of disease symptoms and immune failure. Temperature is an especially compelling mechanism considering that healthy individuals were abundant below depths of 19 metres where the water was cooler (E Lim, pers. comm., 2021). The effect of temperature-related stressors on wasting in sea cucumbers has yet to be tested.

Here, we assess whether acute heat stress rapidly induces wasting in *A. californicus*. We also evaluate behavioural stress response signatures through stiffening, spawning, and evisceration to explore how they are affected by extreme elevated temperature. Sea cucumbers are important benthic detritivores that play an important role in organic matter decomposition and sediment aeration (*Purcell, Conand & Byrne, 2016*). Evidence of a connection between temperature and mass wasting events in sea stars (*Bates, Hilton & Harley, 2009*; *Eisenlord et al., 2016*; *Harvell et al., 2019*) necessitates a proactive assessment of whether heat stress can induce similar symptoms in sea cucumbers.

## MATERIALS & METHODS

### Collection and acclimation

We collected *Apostichopus californicus via* SCUBA from a depth of 7–12 m in July 2021 from Scott's Bay and Bamfield Inlet in Barkley Sound, British Columbia (48°50′02″N, 125°08′45″W). The sea cucumbers were initially collected and used in an unrelated short term tagging experiment comparing T-bar and coded wire tags at the end of July 2021, after which they were all tagged with coded wire tags. Tagging had no measurable impact on sea cucumber behaviour or physiology in the short or long term (Leedham et al., unpublished data; *Cieciel, Pyper & Eckert, 2009*). After this experiment sea cucumbers were then maintained in an open flow-through system at the Bamfield Marine Sciences Centre, with inflow from Bamfield Inlet for four months prior to the start of the thermal stress experiment.

### Experimental design

To determine an average wet weight of each individual, sea cucumbers were weighed twice at the beginning of the experiment. Individuals ($N = 56$) were randomly divided into one of three temperature treatments: control (12 °C, $n = 19$), warm (17 °C, $n = 19$), or heatwave (22 °C, $n = 18$). Our experimental temperatures reflect average summer temperatures and those recorded in the Strait of Georgia when wasting of *A. californicus* was observed (Fig. 1D). Specifically, the average temperature across July and August 2020 and 2021 was 17.5 °C at 5 m depth and 11.9 °C at 20 m depth (Fig. S1). In the summer 2021 heatwave, a maximum temperature at 5 m depth of 21.6 °C was reached on August 04, 2021 (*Pawlowicz, 2017*; *Xuereb et al., 2018*; *Chen et al., 2021*; *Ocean Networks Canada Data Archive, 2022*). Our control temperature (12 °C) was that of the Bamfield Inlet at the time of the temperature experiment and was the temperature the cucumbers were acclimated to.

We separated the sea cucumbers into 27 isolated experimental containers ($61 \times 41 \times 22.2$ cm) with two individuals per container, aside from two smaller containers ($33 \times 45.7 \times 11.4$ cm) that housed one individual each. A plastic mesh divider was used to separate sea cucumbers within each container to allow for tracking individuals throughout the experiment. The containers were randomly placed into sea tables which acted as water baths. The target water bath temperatures were achieved by having a constant flow of cooler inlet sea water to act as the control (12 °C), allowed to reach ambient room temperature for the warm treatment (17 °C), or heated using two 800 W aquarium heaters for the heatwave treatment (22 °C). The 22 °C treatment temperature was gradually increased over a 24-hour period in the sea table water baths (day 1; Fig. S2). During the temperature treatment period, there was no seawater flow to the experimental containers in which sea cucumbers were housed for any of the temperature treatments. To prevent stagnation and maintain high oxygen levels, all containers were constantly aerated with airstones. The water remained at target temperatures for 79 h, after which they were lowered back to the control temperature of 12 °C over 9-hours. Days five through 12 were a recovery period where all sea cucumber holding containers were maintained on the flow through system. Throughout the experiment, the sea cucumbers were closely monitored. To avoid water

fouling, water changes were completed in individual containers as needed to maintain nitrate and ammonium levels below 0.5 ppm, and fresh sea water was heated to the appropriate treatment temperature prior to water changes. Water quality was measured using API® saltwater ammonium and nitrate kits, and food was withheld during the thermal stress window. We monitored if sea cucumbers were defecating three times each day throughout the experiment, based on the presence of fecal pellets, as cessation of defecation is considered to be indicative of a loss or atrophy of digestive organs (*Swan, 1961*; *Fankboner & Cameron, 1985*).

## Phenotyping sea cucumber stress

We assessed sea cucumbers for wasting symptoms throughout the experiment and counted skin lesions on days 5, 6, 7, and 12. Epidermal damage was classified as either minor lesions or major ulcers based on their size and appearance. We defined minor lesions as regions of epidermal damage at the spine tips without signs of considerable discoloration, where the dermis was not damaged across its full thickness (Fig. 2B). We classified major ulcers as epidermal ulceration exposing underlying white mutable connective tissues beneath (Fig. 2D).

We assessed body stiffness using a 3-point ordinal scale to measure heat-stress induced deviations from normal behaviour. To do so, first we removed each individual from their bin and gently palpated them for 10 s to encourage them to stiffen. Then we placed them on an elevated platform to measure their ability to maintain their posture over 5 s. The stiffness testing platform was constructed by cutting a piece of plastic pipe into an eight cm long, eight cm in diameter, and four cm tall section, which we glued onto an eight cm tall block. We assigned a score of 0 if the organism failed to stiffen at all (full droop), a score of 1 if it did not remain stiff for the full 5 s when placed on the platform (partial droop), and a score of 2 if it maintained its posture for the entire 5 s (no droop). Stiffness was measured on days 1–5 of the experiment (as a baseline and throughout the treatment), on day 7 (48 h after experimental endpoint) and day 12 (7 days after experimental endpoint). Frequency of handling individuals occurred equally across treatments.

As some sea cucumbers spawned, we recorded whether there was any evidence of spawning every 12 h. Spawning was assigned to each container rather than individual, since some individuals were co-housed ($N_{bin} = 30$). We also evaluated whether specimens had eviscerated based on the presence or absence of ejected internal organs during these routine checks.

## Statistical analyses

All statistical analyses were conducted in R (v4.0.3, *R Core Team, 2020*). We tested the distribution of the number of lesions using the *fitDist* function (*Rigby & Stasinopoulos, 2005*) and determined that minor lesion count best fit a geometric distribution. We modelled the maximum number of minor lesions as a function of treatment, weight, evisceration (binary), and defecation status (binary). We included the sea table and bin as random effects. We then used backwards selection with the *stepGAIC* function from GAMLSS (*Rigby & Stasinopoulos, 2005*) to determine the most parsimonious combination of variables that best explained the maximum number of lesions.

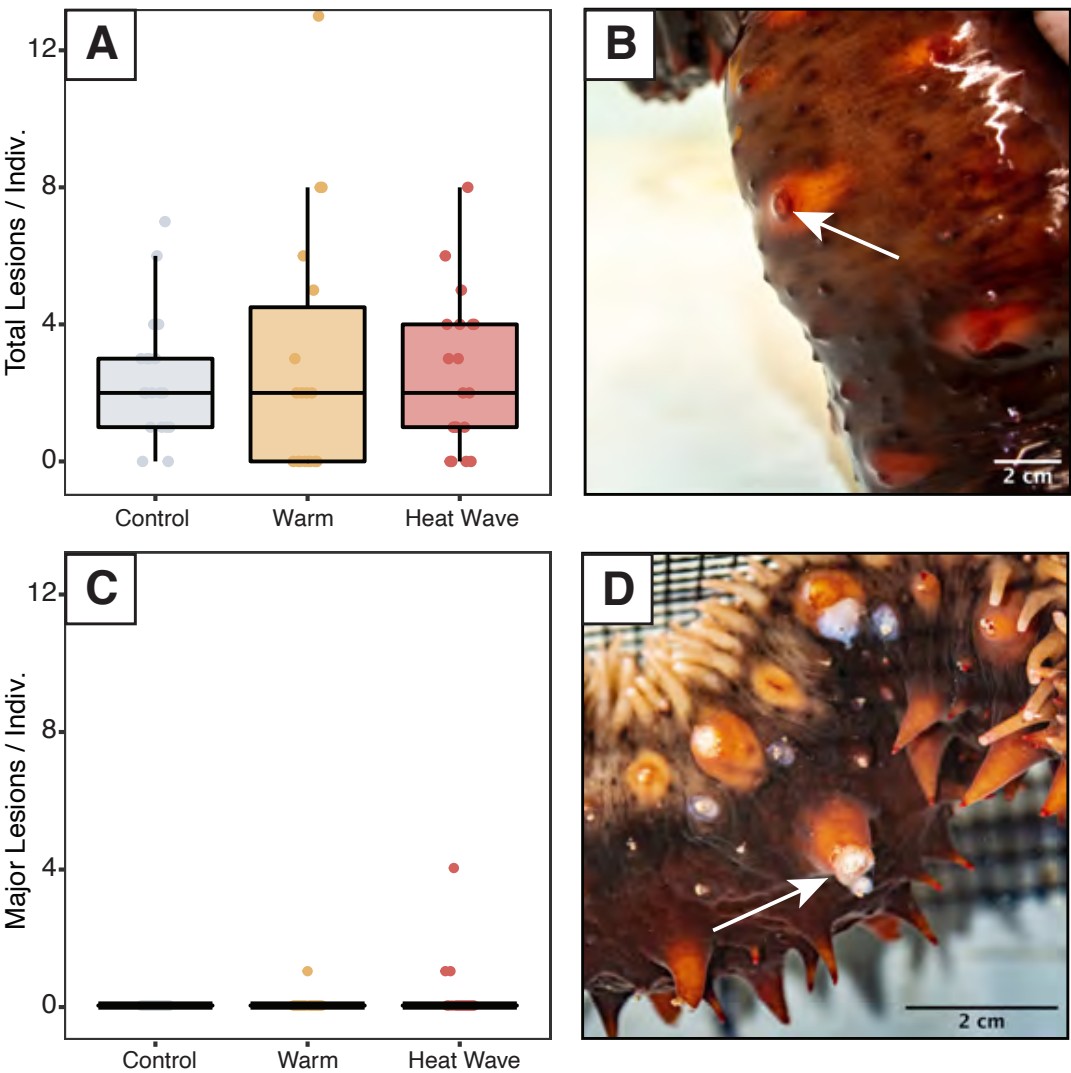

**Figure 2** **Skin ulcers observed in *Apostichopus californicus* across treatments.** (A) Minor ulcer count, defined as small lesions on the ends of spines (B). (C) Major ulcer count, defined as open wounds that expose white subdermal tissue (D).

We examined the covariates that affected the likelihood of mortality with a logistic regression model. Sea cucumber mortality was a binary measure, with treatment, evisceration, defecation status, initial droop score, and initial weight as explanatory variables. Terms for the final model were selected *via* backwards model selection using *stepGAIC* (*Rigby & Stasinopoulos, 2005*) to produce the most parsimonious model of variation in sea cucumber mortality.

To assess changes in stiffness, we constructed full ordinal regression models using the *clmm* function (*Christensen, 2019*) with temperature treatment, date, and the interaction between treatment and date as explanatory variables. We restricted stiffness measurements to those taken before the heat treatment began (day 1), during the treatment (days 2–4)

and immediately after the heat treatment (day 5). We included individual identity as a random effect to account for repeated measures on the same individuals over time. We also included bin and sea table as random effects to account for the paired (co-housing) and blocked (five bins per sea table) experimental design. We used backwards model selection to determine the most parsimonious models using *dredge* (*Bartoń, 2020*).

To assess the drivers of evisceration, we constructed a logistic regression model with treatment, weight, and defecation status as explanatory variables, along with sea table as a random effect. We determined the most parsimonious model through backwards selection. To compare the incidence of stress spawning across temperature treatments, we conducted a Dunn's Kruskal-Wallace (K-W) test using the *dunnTest* function from the FSA package (*Ogle et al., 2022*).

All data and annotated code for the analyses described above are publicly available at https://github.com/declan-taylor/sea_cucumber_wasting (DOI: 10.5281/zenodo.7259325).

# RESULTS

The temperature treatments varied slightly from the target temperatures during the experimental heatwaves. During the 79 h treatment, the mean temperature of the control treatment was 12.4 °C and varied from 10.8 - 14.0 °C; the mean of the warm treatment was 16.6 °C and ranged from 14.8 °C to 17.9 °C; the mean of the heatwave treatment was 21.7 °C and varied from 19.6 °C to 23.3 °C. Temperature treatments were significantly different from each other (K-W $\chi 2 = 463.32$, $df = 2$, $p < 2.2e{-}16$).

Minor skin lesions occurred during the experiment and were not statistically different between treatments ($n_{control} = 17$, $n_{warm} = 15$, $n_{heatwave} = 10$; Fig. 2). Major ulcers were only observed in the warm ($n = 1$) and heatwave treatments ($n = 2$). These major ulcers appeared to heal throughout the recovery period and were re-classified as minor lesions on day 12. The maximum number of minor lesions per individual was not significantly explained by treatment, weight, evisceration, or defecation status.

Mortality was only observed in the 22 °C heatwave treatment (Fig. 3; $n = 5$), while there were no mortalities observed in the control or warm treatments. Based on backwards model selection, mortality was driven by treatment and weight (Table S1). Body stiffness was lower in the warm and heatwave treatments compared to the control treatment (Fig. 3). Backwards-selected models indicated that temperature treatment and day affected stiffness (Table S2). Sea cucumbers were significantly less likely to have high stiffness scores relative to the control treatment in the warm ($p = 1.99e{-}7$) and heatwave ($p = 2.44e{-}11$) treatments. Structural stiffness values were significantly likely to be lower than day 1 on day 3 ($p = 1.37e{-}5$), day 4 ($p = 2.50e{-}5$) and day 5 ($p = 8.66e{-}5$), but not on day 2 ($p = 0.0627$; Table S2).

Stress spawning was observed during the temperature treatment ($n = 11$ bins), with spawning occurring in the 17 °C warm ($n = 5$ bins) and the 22 °C heatwave ($n = 4$ bins) treatments. However, there was no significant difference in spawning between temperature treatments (K-W $\chi 2 = 1.94$, $df = 2$, p = 0.379).

Evisceration was observed across all treatments ($n_{control} = 2$, $n_{warm} = 5$, $n_{heatwave} = 5$). Treatment temperature did not explain a significant amount of the variance in the

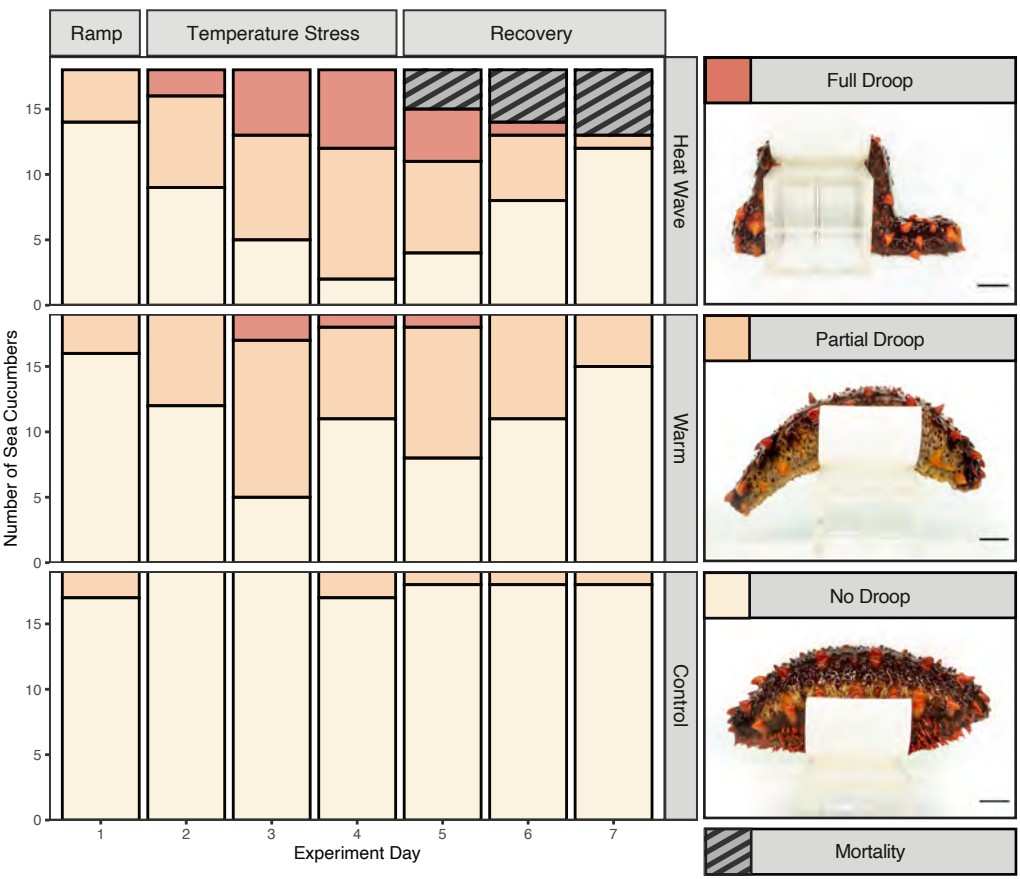

**Figure 3  Stiffness and mortality of *Apostichopus californicus*.** Heat ramp (day 1), during temperature stress (days 2–4), and recovery (days 5–7) from the temperature treatment.

occurrence of evisceration (Table S3), however, weight ($p = 0.0383$) and defecation status ($p = 0.0163$) were significant drivers increasing likelihood of evisceration (Table S3). Along with evisceration of internal organs, we also observed the expulsion of the respiratory tree in two individuals. Both individuals were in the 22 °C heatwave treatment and mortality shortly followed the expulsion of the respiratory tree.

## DISCUSSION

The objective of our study was to determine if heat stress can induce wasting symptoms in *Apostichopus californicus*. While we saw minor skin lesions at all treatment levels, and major ulcers in the warm and heatwave treatments, these were not characteristic of wasting symptoms observed previously (*Hewson, Johnson & Tibbetts, 2020*; Fig. 1). Neither the minor lesions nor major ulcers that we observed matched the wasting symptoms reported in *A. californicus* in Nanoose Bay, B.C. (E Lim, pers. comm., 2021), or the isolated wasting events reported along the Pacific coast (*Hewson, Johnson & Tibbetts, 2020*). We also did not see any sloughing of body tissues or liquefaction, as has been anecdotally reported in previous wasting events (*Schroeder, 2017*; *Hewson, Johnson & Tibbetts, 2020*). Despite

the major ulcers sharing some resemblance in colour, texture, and location to wasting disease symptoms, the sea cucumbers did not exhibit the full suite of symptoms that are typical from a wasting cucumber, and, within the treatment period, these lesions did not develop into fissures covering the dorsal body wall as seen in the wild (Fig. 1C). In Nanoose, wasting symptoms developed into a mass outbreak and mortality event over several months. However, unlike in reports of widespread wasting-driven mortality in wild *A. californicus*, the major ulcers in our specimens healed within the 7-day recovery period, and there was no evidence of these symptoms persisting or spreading to co-housed individuals. As such, we could not conclude that the sea cucumbers in our experiment were afflicted by the fatal wasting disease that has been previously reported (*Schroeder, 2017*; *Hewson, Johnson & Tibbetts, 2020*). Future experiments could address the multi-week delay between the heat dome and maximum die-off through prolonged exposure to elevated temperatures and utilise histopathology to further investigate the etiology of epidermal ulceration.

While we are uncertain of the ultimate cause of the lesions that we observed, they were likely associated with handling while assessing their posture maintenance, which increased the frequency of potential abrasions on the epidermal surface tissue. Major ulcers may have begun as minor lesions that were then exacerbated by the high physiological stress caused by the 22 °C treatment. White skin ulcerations, like the major ulcers we observed, are a recognized condition in other Holothuroidea, described as a Skin Ulceration Disease or Skin Ulceration Syndrome (SUS; *Delroisse et al., 2020*). SUS has been documented in commercially farmed *Apostichopus japonicus* and *Holothuria scabra*, and has been characterized by white ulcers on both sides of the body wall (*Wang et al., 2007*; *Deng et al., 2009*; *Li et al., 2012*; *Zhang et al., 2018*). Minor SUS symptoms in commercially farmed *A. japonicus* and the major ulcers in our *A. californicus* specimens are visually similar (*Deng et al., 2009*; *Zhang et al., 2018*). Unlike the SUS symptoms reported in *A. japonicus*, we did not see any indication of swelling or discolouration of the peristomes, and we did not see an initial abundance of ulcers around the mouth or cloaca (*Becker et al., 2004*; *Wang et al., 2007*; *Delroisse et al., 2020*). Extreme cases of SUS in farmed *A. japonicus* also bear resemblance to the wasting symptoms in wild *A. californicus*, raising further questions about the causes of skin ulceration in Holothuroidea (*Delroisse et al., 2020*).

Despite not seeing evidence of wasting, we observed compelling evidence of stress symptoms in response to thermal stress. We observed five mortality events in the 22 °C heatwave treatment. None of these mortalities were co-housed together, suggesting that it is unlikely a fatal disease spread between cohoused individuals. Therefore, we suggest that the heatwave treatment is close to the upper critical thermal tolerance of *A. californicus*, but the warm treatment temperature does not cause sufficient stress for mortality to occur. Our findings align with previous work on larval life stages, where *Ren, Liu & Pearce (2018)* found that at 22 °C, larval *A. californicus* experienced reduced survival which was not observed at 16 °C or 18 °C.

Beyond lethal effects, we observed reduced stiffening behaviour associated with both the 22 °C heatwave and 17 °C warm water treatment. A temperature-induced loss in stiffness may have implications for sea cucumber fitness under warming sea temperatures,

as a limited ability to stiffen may inhibit their ability to avoid predation or maintain posture while feeding and spawning. Thermal stress may have reduced stiffness by causing muscular fatigue and relaxation (*Dowd & Somero, 2013*) in the circular and longitudinal-ambulacral muscles (*Gao & Yang, 2015*). Stiffening is also caused by protein-mediated changes in mutable collagenous tissue within the dermis of sea cucumbers (*Takehana et al., 2014*). Therefore, heat stress may have reduced stiffening by denaturing or decreasing the production of tensilin, a stiffening protein, or increasing the production of the de-stiffening protein softenin (*Yamada et al., 2010*; *Takehana et al., 2014*; *Tamori et al., 2016*).

Unlike stiffening behaviour, we did not observe any significant trends in spawning behaviour or evisceration. Of the 11 spawning events, nine occurred in elevated-temperature treatments (warm or heatwave) and this trend, although statistically insignificant, was expected as stress spawning has been previously reported in other sea cucumber species (*Battaglene et al., 2002*; *Rakaj et al., 2018*; *Schagerström et al., 2021*). We were unable to associate spawning with individual phenotypes due to the co-housing experimental design. Evisceration appeared random across treatments, potentially because *A. californicus* in all treatments may have been reacting to handling stimulation and stress during the experiment (*Ding et al., 2019*). However, we found evidence that biological mechanisms, weight and defecation status, partially explain the non-treatment related variation in evisceration (Table S1). Defecation status in particular showed that *A. californicus* that were not defecating were more likely to eviscerate. This may have occurred because the energetic cost of eviscerating digestive organs would be lower for *A. californicus* that had already ceased using their organs, either because they were preparing to eviscerate (*Swan, 1961*) or undergoing viscera atrophy (*Fankboner & Cameron, 1985*). As such, when stressed by handling, *A. californicus* that had already begun seasonal reductions in digestive function may have been more likely to eviscerate from stress and overstimulation. Unlike digestive tract evisceration, we do not believe that the expulsion of the respiratory tree in the 22 °C heatwave treatment was linked to seasonal senescence. For both individuals, respiratory evisceration was followed by mortality, suggesting that this is an indication of extreme physiological distress from thermal exposure.

Since we observed skin lesions under thermal stress that were likely unassociated with wasting, other factors are likely causing recent wasting outbreaks in *A. californicus*. Like wasting, severe cases of SUS are highly transmissible and result in mortality (*Delroisse et al., 2020*); bacterial and viral pathogenic causal agents have been previously linked to SUS and wasting-like symptoms, both in other sea cucumbers (*Deng et al., 2008*; *Deng et al., 2009*; *Liu et al., 2010*; *Zhang et al., 2018*; *Delroisse et al., 2020*) and sea stars (*Hewson et al., 2014*; *Hewson et al., 2018*; *Work et al., 2021*). A study examining a single wasting *A. californicus* specimen found a high viral load but was unable to identify a specific pathogen causing wasting symptoms (*Hewson, Johnson & Tibbetts, 2020*). Avenues for future research on wasting diseases in *A. californicus* could address the potential for shared etiology with SUS, given the symptomatic similarities, as well as the possibility of pathogenic origins. Such investigations would benefit from histopathological examinations of potentially diseased tissue to reveal more about the symptom's pathogenesis. Infectious factors (viral, bacterial) and non-infectious factors (chemical pollution, hypoxia, eutrophication) should both be

investigated because widespread environmental degradation and anthropogenic climate change are shifting pathogenic dynamics globally (*Marcogliese, 2008*; *Allan et al., 2021*). These investigations would add valuable insight to the field of wasting diseases, given the scarcity of published information on these events in echinoderms.

In this study, we exposed *A. californicus* to extreme thermal stress as measured by mortality, degraded stiffening behaviour, and the development of skin lesions. Despite this, we found no evidence that wasting is induced by temperature stress alone. Therefore, the August 2021 mass wasting event in Nanoose, British Columbia, was likely not triggered solely by the anomalous heatwave. Determining the factors that cause and exacerbate wasting in *A. californicus* is essential for predicting and managing mass mortality events. Sea cucumbers are ecologically important benthic detritivores, which break down organic matter, recycle nutrients, and maintain sediment health (*Wheeling, Verde & Nestler, 2007*; *Purcell, Conand & Byrne, 2016*). Efforts to protect, manage, and sustainably harvest *A. californicus* in the face of global environmental change will require a comprehensive understanding of their stress responses, disease dynamics, and the novel threat of sea cucumber wasting.

## ACKNOWLEDGEMENTS

The species collections and experiments took place on the traditional territories of the Huu-ay-aht First Nations, a Nuu-chah-nulth Nation and signatory to the Maa-nulth First Nations Final Agreement, and we are grateful for the opportunity to conduct research in protected and sacred areas. We would like to thank Chloe Curry, Arya Horon, and Juliane Jones for providing lab assistance; Payton Arthur, Mike Chung, Gabrielle Languedoc, Sammie Foley, Juliane Jones, and Carter Burtlake for feedback on early versions of this manuscript.

### Funding
Funding was provided by Bamfield Marine Sciences Centre to the Fall Program and by Mount Royal University to Sara J Smith Wuitchik. The funders had no role in study design, data collection and analysis, decision to publish, or preparation of the manuscript.

### Grant Disclosures
The following grant information was disclosed by the authors:
Bamfield Marine Sciences Centre to the Fall Program.
Mount Royal University.

### Competing Interests
The authors declare there are no competing interests.

## Author Contributions

- Declan Dawson Taylor conceived and designed the experiments, performed the experiments, analyzed the data, prepared figures and/or tables, authored or reviewed drafts of the article, and approved the final draft.
- Jonathan J. Farr conceived and designed the experiments, performed the experiments, analyzed the data, prepared figures and/or tables, authored or reviewed drafts of the article, and approved the final draft.
- Em G. Lim conceived and designed the experiments, prepared figures and/or tables, and approved the final draft.
- Jenna L. Fleet conceived and designed the experiments, prepared figures and/or tables, and approved the final draft.
- Sara J. Smith conceived and designed the experiments, analyzed the data, prepared figures and/or tables, authored or reviewed drafts of the article, and approved the final draft.
- Daniel M. Wuitchik conceived and designed the experiments, analyzed the data, prepared figures and/or tables, authored or reviewed drafts of the article, and approved the final draft.

## Data Availability

The data and code is available at GitHub: https://github.com/declan-taylor/sea_cucumber_wasting.

## Supplemental Information

Supplemental information for this article can be found online at http://dx.doi.org/10.7717/peerj.14548#supplemental-information.

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
