# Peer review of "Heat stress does not induce wasting symptoms in the giant California sea cucumber (Apostichopus californicus)"

_PeerJ, doi:10.7717/peerj.14548_

## Round 0.1 · original submission · Major Revisions

Both reviewers have included a number of useful comments that will improve the manuscript. Both reviewers suggested stronger links between the observations and lab work, and noticed a few minor things (like missing Figure 4).

·

Basic reporting

Heat stress does not induce wasting symptoms in the giant California sea cucumber (Apostichopus californicus) by Declan Taylor Equal first author, 1, 2 , Jonathan J Farr Equal first author, 2, 3 , Em G Lim 4 , Jenna L Fleet 2, 5 , Sara J Smith Wuitchik, Daniel M Wuitchik PeerJ #74405

Overview: Hypothesizing that cucumber wasting disease (CWD) is temperature induced, authors attempt to replicate disease by exposing cucs to elevated temperature in captivity to replicate gross lesions seen in the wild. Although they do reproduce focal ulceration and mortality in cucs exposed to high temperatures, the lack of congruence of gross lesions in captive vs wild cucs lead them to conclude that elevated temperature was likely not the incriminating factor leading to CWD. Although there is likely some useful information in this MS, the authors need to think the problem through a bit more before this is publishable or before they can make such a conclusion. Several points: 1) CWD is a poorly defined disease. Given that it is mainly described based on gross lesions (presence of severe epidermal ulceration in cucs), there are myriad things that could lead to that and absent additional diagnostics such as doing histopathology on affected cucs to better define epidermal lesions, the case definition for this disease will continue to be vague. 2) It is very difficult to disentangle effects of husbandry vs treatment (temperature) as actual precipitating causes of epidermal lesions in captive cucs. Authors need to do a better job convincing reader that animals did not develop lesion simply because of handling, housing, feeding, etc. Captive experiments in echinoderms are fraught with difficulties because these animals can often develop lesions simply because of the husbandry conditions they are undergoing. 3) The data analyses do not make sense. You subjected animals to 3 temps. So, it would seem that doing a multiple regression with all the variables you measure (weight loss, lesion severity, defecation, etc) vs temp might be more reasonable. Moreover, I suspect you are missing an opportunity to shed light on pathogenesis of epidermal ulceration by not analyzing data temporally. What follows are specific comments in attempts to help things along, hopefully in the right direction.

Line 139: It would be useful here to get mean length and diameter of cucs to give a reader an idea of housing density. Ideally, some sort of index such as estimated percent of volume of container occupied by cucs. This seems important to 1) be sure that somehow excessively high density of cucs or water fouling were not an issue and 2) that housing density was comparable between treatment groups (to ensure there was not some sort of management effect artefact in experimental design). I assume 9 animals per temperature treatment? Size of animals was comparable for all treatments?

Line 144: It would be useful here to have flow rate and water turnover in tanks (e.g. X volumes changes/h).

Line 147: OK, here it might be difficult to sort out whether the issue is water flow or temperature no? If you stopped water flow to heat cucs, did you also stop water flow for controls? Same applies for ambient. Perhaps the issue in the experimental design is an interaction between water flow and temperature?

Line 148: Why was this particular time period (79h) chosen?

Line 154: What kind of food was offered and how often? I'm assuming tank fouling was not an issue? The sentence on monitoring defecation would go best here. How often was defecation monitored? Was water quality monitored? Just presence or absence or did you quantify fecal pellets? Did you weigh or measure the animals at start and end of experiment? Might that have given valuable data on body condition?

Line 157: I assume here wasting symptoms refer to epidermal ulceration correct? If so, why not so state? What made you pick those particular dates for monitoring ulcers? I suppose this was because it was after treatment? I assume that there was a Time 0 assessment for skin ulcers? Were ulcers counted dorsal and ventral?

Line 160: I believe you mean epidermis here. Actually, ulcer is a very specific term indicating loss of epidermal layers. This is obvious when you see white tissue. However, your small lesions may have simply been epidermal thinning. You really can't tell without doing histopathology (too bad that wasn't done....might have yielded valuable info on pathogenesis of epidermal ulceration in cucs). I would suggest to be more anatomically correct that you relabel as minor: multifocal epidermal discoloration of spine tips and Major: Epidermal ulceration exposing underlying white mutable connective tissues.

Line 165: I really like this platform system. Very creative way to measure mutable connective tissue stiffness. Please refer reader to the appropriate figure as illustration. Also, pls provide dimensions of platform. This could be a very useful and simple tool to investigate cuc health. Why 5 seconds monitoring? Based on pilot studies showing they soften within that time?

Line 175: Some individuals were co-housed. Hm. This bring into question the experimental design and housing as an experimental confounder. Please see my comments above regarding housing density and water circulation. You really need to convince the reader here that your results are not an artefact of housing conditions vs water flow vs temperature.

Line 181: What about major ulcers? How did that fit in the analysis? Did you consider adopting an overall ulcer severity score (e.g. 0, 1, 2, or 3 depending on whether there were no, mild, moderate or severe ulceration) and doing your regressions based on that? You could use that severity score as part of the covariates to explain mortality and eliminate the need for the analysis described in lines 181-186. That analysis does not make much sense to me anyway. Why would weight, evisceration, and defecation explain ulcers and not the other way around? Does it not make more biological sense that ulcers would lead to inappetance, lower defecation, and more evisceration? It seems the analysis starting in line 188 makes more sense in that that is your central question: Does temperature lead to ulceration, loss of MCT stiffness, mortality, defecation, and evisceration? And if you are interested in teasing out the clinical signs, then it would seem that time would be an important factor here. As such, perhaps plotting when clinical signs arise over time might be very informative as to pathogenesis of heat stress in cucs. For instance, in SSWD, it is hypothesized based on histopath, that MCT collapses first followed by epidermal ulceration. Your nifty stiffness test might show this nicely (e.g. MCT goes soft then ulcers appear).

Line 204-210: Consider reformulating your analyses based on above. This particular analysis does not make much biological sense to me either.

Lines 244-245: I was unable to find Figure 4.

Line 250-266: I'm not sure I agree with this. Wasting symptoms is just another form of ulceration, albeit much more severe than what you saw here. That is a pretty broad and non-specific case definition. Indeed I would bet in the wild, wasting disease in cucs likely has a range of manifestations from mild to severe (as in Fig. 1). Who's to say that mild wasting disease does not look like what you saw here. In fact, I would bet during field surveys that mildly affected cucs may go undetected. The best conclusion you can draw is that exposing urchins to 22C for X h does not lead to severe epidermal ulceration. Perhaps if you exposed them for longer, ulcers would grow more severe. Very hard to say. I would definitely tone down this paragraph.

lines 268-284: OK, the management issue is again rearing its head (see above). Were animals by chance handled equally frequently? If not, that is another big red flag confounder is it not, as you correctly acknowledge? But really hard for reader to judge without more detail on housing and management. Has anyone in these cited papers done any additional laboratory diagnostics to sort out why cucs might be ulcerating (e.g. hematology, histopathology)?

Line 289: Infectious disease. Hard to say. Your experimental design really was not suitable to address that. Had you examined tissues for histopathology, you might be in a better place to make that argument.

Lines 296-306: This is probably the most creative part of the study. See my comments above on temporal analysis of clinical signs which might shed light on pathogenesis of temperature-induced ulceration in cucs.

Lines 308-328: Did you weigh animals at beginning and end of experiment? Seems that would give very valuable data on body condition, eating, defecation in that it would integrate all those signals in a single metric. For instance, was weight loss most severe in high temp vs control? If you have percent weight loss, this could also go into your regression analysis looking at variables vs temperature.

Line 337: Shared pathology with SUS...what do you mean by that? Gross pathology? Has anyone done histopath? Seems you could recommend something more substantial such as "Future investigations such as these might benefit from more systematic biomedical examinations including examination of blood and tissues under the microscope to shed light on pathogenesis of condition."

Lines 342-351: Delete. Pure speculation.

References

Does the Christensen 2019 reference have any other info (publisher?, pages?, editor?) Same with Ogle 2021

Figure S1. What do the line colors represent?

Experimental design

See Basic Reporting

Validity of the findings

See Basic Reporting

Additional comments

See Basic Reporting

Reviewer 2 ·

Basic reporting

This manuscript is well written, well cited and addresses both interesting and pressing issues. My version of the manuscript appears to be missing one figure (Figure 4).

Discussed in more detail in the line edits, but I do think there is some work needed to address the logical link between the field observations and the lab work. The lab experiments show that short term exposure and immediately following exposure to high temperature the A. californicus don’t show signs of wasting. However, in the field there was a substantial delay between the heat-wave event and the observation of the wasting event. I think implications of what that delay might mean could be discussed in the introduction – e.g. what does it tell us about the implied mechanistic link between the wasting event and the heat-wave? I also think it should be discussed in the context of the discussion – e.g. what are the limitations to only monitoring for a week following the lab-induced heat-wave? I still find this work very relevant and interesting, and I think having these additional framing points will help to place the new information in the best light.

Please find some line edits on the introduction below:

Line 32-34: While this sentence is technically correct, fast reading of the current structure leaves the reader with the impression that you are implying that the heatwave ‘resulted’ in the mortality. I would re-order the sentence to help to clarify that you are implying that the wasting event resulted in mortality, all of which came after the heat-wave.

Line 81: I would remove the aside ‘similar to sea star wasting disease’. While bacteria are implicated in sea star wasting disease, there remains much debate in the literature on what the causative agent (s) are, as you state earlier in your introduction (line 64-71).

Line 82-84: This sentence needs citations.

Line 93-101: I think that a bit more time should be spent discussion the implications of the timing of the temperature event and the wasting outbreak. What does the delay between the extreme temperature and the response imply?

Experimental design

The design and statistical analysis were appropriate to the question and well designed. I had a few specific points of interest, in line edits below:

Line 117-119 and line 128-130: It would be good to note whether there were any ‘treatments’ from this previous experiment, and how the previous treatment was handled in the distribution of individuals between this experiment’s treatments. In other words, did you control for the previous experiment’s effects in the placement of the animals in your temperature treatments? Even though you state that the effects were minor from the previous experiment, it would be good to know if there could be any underlying patterns within your temperature treatments.

Line 144-148: Did you have aeration or water pumps in the 17 and 22 degree treatments to keep water movement and oxygen levels high? If not, there could be substantial differences in oxygen availability between the control with its flowing sea water, and the two temperature treatments.

Validity of the findings

As discussed above, I think there needs to be more work relating your lab experiment to the field observations. Specific points in line edits below:

Line 25-256 and 355-357: I think that you haven’t addressed the timing issues with your experiment. The heat-wave in the wild happened in June/July, and the wasting event was observed from August-October. If we assume that the heat-wave precipitated the wasting event in the field, then there was at least a 1-to-2-month delay between the onset of the driver and the response. I think you need to discuss this in both your introduction or discussion. I still think that the work is interesting on its own right, but if you are trying to invoke the same mechanism as what was found in the field, then you would need to monitor your lab experiment for substantially longer.

---

## Round 0.2 · Minor Revisions

The revision is much improved -- the reviewer has a couple of lingering things, which should be easy to address.

·

Basic reporting

Heat stress does not induce wasting symptoms in the giant California sea cucumber (Apostichopus californicus) by Declan Taylor Equal first author, 1, 2 , Jonathan J Farr Equal first author, 2, 3 , Em G Lim 4 , Jenna L Fleet 2, 5 , Sara J Smith Wuitchik, Daniel M Wuitchik PeerJ #74405

Overview: This is a revised MS. The authors have done a nice job addressing reviewer comments and revising the MS. I only have a few more minor edits to polish things up. Well done!.

Line 158: Pls indicate methods used to measure nitrates and ammonium

Lines 35-359: I would stay away from terms like 'biotic' and 'abiotic' when you are referring to disease or lesions in animals (both of which are biotic processes). A more logical distinction is 'infectious' and 'non-infectious' processes.


Figure 2. A) "Minor ulcer; note ill-defined area of integument pallor (arrow)." D) "Major ulcer; note complete absence of epidermis revealing underlying white mutable connective tissue (arrow)."

Experimental design

See above

Validity of the findings

See above

Additional comments

See above

---

## Round 0.3 · accepted · Accept

Thanks for making this final round of changes, and congratulations!